# CXPlain: Causal Explanations for Model Interpretation under Uncertainty

**Patrick Schwab and Walter Karlen**
Institute of Robotics and Intelligent Systems, ETH Zurich
`patrick.schwab@hest.ethz.ch`

## Abstract

Feature importance estimates that inform users about the degree to which given inputs influence the output of a predictive model are crucial for understanding, validating, and interpreting machine-learning models. However, providing fast and accurate estimates of feature importance for high-dimensional data, and quantifying the uncertainty of such estimates remain open challenges. Here, we frame the task of providing explanations for the decisions of machine-learning models as a causal learning task, and train causal explanation (CXPlain) models that learn to estimate to what degree certain inputs cause outputs in another machine-learning model. CXPlain can, once trained, be used to explain the target model in little time, and enables the quantification of the uncertainty associated with its feature importance estimates via bootstrap ensembling. We present experiments that demonstrate that CXPlain is significantly more accurate and faster than existing model-agnostic methods for estimating feature importance. In addition, we confirm that the uncertainty estimates provided by CXPlain ensembles are strongly correlated with their ability to accurately estimate feature importance on held-out data.

## 1   Introduction

Explanation methods for machine-learning models play an important role in researching, developing, and using predictive models as information on what features were important for a given output enable us to better understand, validate, and interpret model decisions [1–5]. However, complex models, such as ensemble models and deep neural networks, are often difficult to interrogate. To address this apparent dichotomy between performance and interpretability [6], researchers have developed a number of attribution methods that provide estimates of the importance of input features towards a model's output for specific types of models [4, 7–15], and for any machine-learning model [6, 16].

However, providing fast and accurate feature importance estimates for any machine-learning model is challenging because there exists a wide variety of intricate machine-learning models with different underlying model structures, algorithms, and decision functions, which makes it difficult to develop an optimised and unified approach to importance attribution. Furthermore, importance estimates of state-of-the-art methods are typically associated with significant uncertainty [3, 17–19], and it is therefore difficult for users to judge when importance estimates can be expected to be accurate.

In this work, we present a new approach to estimating feature importance for any machine-learning model using causal explanation (CXPlain) models. CXPlain uses a causal objective to train a supervised model to learn to explain another machine-learning model. This approach can be applied to any machine-learning model, since it has no requirements on the predictive model to be explained. In particular, it does not require retraining or adapting the original model. We demonstrate experimentally that CXPlain is significantly more accurate than most existing methods, fast, and able to produce accurate uncertainty estimates. Source code is available at `https://github.com/d909b/cxplain`.

**Contributions.** This work contains the following contributions:

- We introduce causal explanation (CXPlain) models, a new method for learning to accurately estimate feature importance for any machine-learning model.
- We present a methodology based on bootstrap resampling for deriving uncertainty estimates for the feature importance scores provided by CXPlain.
- Our experiments show that CXPlain is significantly more accurate and significantly faster (at evaluation time) than existing model-agnostic methods, and that the uncertainty estimates for its assigned feature importance scores are strongly correlated with the accuracy of the provided importance scores on previously unseen test data.

## 2 Related Work

**Feature Importance Estimation.** Existing methods for feature importance estimation can be subdivided into (1) gradient-based methods, (2) methods based on sensitivity analysis, (3) methods that measure the change in model confidence when removing input features, and (4) mimic models. Simple Gradient (SG) [8], Integrated Gradients (IG) [10], DeepLIFT [1], and DeepSHAP [6] are examples of gradient-based methods. Gradient-based methods are only applicable to differentiable models, such as neural networks, and their computation is typically fast. Methods that quantify a model's sensitivity to changes in the input, such as LIME [16] or SHAP [6], and more specifically Kernel SHAP, are applicable to any machine-learning model but typically slow to compute, as large numbers of model evaluations are necessary to assess a model's sensitivity. Methods based on masking parts of the input and measuring the model's resulting change in confidence [20] include conditional multivariate models for visualising deep neural networks [21], analysing the effects of erasing parts of their representations [22], image interpretation by identifying the regions for which the model most strongly responds to perturbations [23], and image masking models trained to manipulate the outputs of a predictive model by occluding parts of the input [24]. The fourth main category of approaches to explaining model decisions is to train interpretable models that mimic the decisions of a black-box model that we wish to explain. Tree- [25–27] and rule-based [28] models have been used as mimic models. However, mimic models are not guaranteed to match the behavior of the original model. Besides these four established categories of feature importance estimation methods, structural causal models (SCMs) [29] and Deep Taylor Decomposition (DTD) [30] have also recently been proposed as explanation methods. However, these methods are designed for specific types of models. In addition, the L2X method that uses a variational approximation of mutual information [31] and Bayesian nonparametrics [32] have been proposed to explain a target model. Tsang et al. [33] detected statistical interactions by interpreting the weights learned in neural networks. Beyond feature attribution, testing with concept activation vectors (TCAV) [34] was proposed to visualise the internal state of deep learning models, and influence functions [35] have been used to identify the training data most responsible for a given model decision. A major limitation of most existing methods for feature importance estimation is that they do not inform users when their estimates are significantly uncertain and can not be expected to be accurate.

**Uncertainty and Reliability of Explanations.** Although reliability is necessary for model explanations to be trustworthy, relatively few studies have been concerned with quantifying the uncertainty and robustness of explanation methods. For example, it has been shown that multiple importance estimation methods incorrectly attribute when a constant vector shift is applied to the input [3], that the attributions provided by interpretation methods may themselves contain significant uncertainty [18], that some explanation methods are independent of both the model and the data-generating process and, thus, can not be relied upon for important interpretation tasks [17], and that imperceptibly small perturbations of the input can significantly alter the explanations provided by state-of-the-art

Table 1: Comparison of CXPlain to several representative methods for feature importance estimation.

|  | CXPlain | SG [8] / IG [10] | DeepSHAP [1, 6] | LIME [16] | SHAP [6] |
|---|---|---|---|---|---|
| Accuracy | high | moderate | high | high | high |
| Model-agnostic | ✓ | ✗ | ✗ | ✓ | ✓ |
| Uncertainty estimates | ✓ | ✗ | ✗ | ✗ | ✗ |
| Computation time | fast | fast | fast | slow | slow |

explanation methods without changing the explained model's prediction [19]. These studies highlight the importance of informing users when a given explanation is uncertain and should be discounted.

In contrast to existing works, CXPlain is an explanation model trained with a causal objective to learn to explain the decisions of any machine-learning model without the need to retrain, adapt, or have in-depth knowledge of the explained model. To the best of our knowledge, CXPlain is the first feature importance estimation method that is simultaneously (1) significantly more accurate than most existing methods, (2) compatible with any machine-learning model and data modality, (3) able to provide uncertainty estimates via bootstrap resampling, and (4) fast at evaluation time (Table 1).

## 3  Methodology

**Problem Setting.**  We consider a setting in which we are given a predictive model $\hat{f}$ which processes inputs $X$ consisting of $p$ input features, or groups of features, $x_i$ with $i \in [0 \mathinner{\ldotp\ldotp} p-1]$ to produce outputs $\hat{y} \in \mathbb{R}^k$ of any dimensionality $k$. The predictive model $\hat{f}$ is scored according to an objective function $\mathcal{L} : y \times \hat{y} \to s$ that computes a scalar loss $s \in \mathbb{R}$ after comparing the model's predictive output $\hat{y}$ to a ground-truth output $y \in \mathbb{R}^k$. The mean squared error (MSE) for regression models and the categorical crossentropy for classification models are commonly used examples of such objectives. We note that we specifically do not require access to, or knowledge of, the process by which $\hat{f}$ produces its output, nor do we require $\hat{f}$ to be differentiable or of any specific form. Additionally, we are given $N \in \mathbb{N}$ independent and identically distributed (i.i.d.) pairs of sample covariates $X$ and ground-truth outputs $y$ as training data. Given this setting, our goal is to train an explanation model $\hat{f}_{\exp}$ that produces accurate estimates $\hat{A}$ with elements $\hat{a}_i$ corresponding to the importances assigned to each of the $p$ input features $x_i$ to the predictive model $\hat{f}$.

**Causal Explanations (CXPlain).**  The main idea behind CXPlain is to train a separate explanation model $\hat{f}_{\exp}$ to explain the predictive model $\hat{f}$ (Figure 1). This flexible framework has the advantage that we do not need to retrain or adapt the predictive model $\hat{f}$ to explain its decisions. To train the explanation model, we utilise a causal objective function that quantifies the marginal contribution of either a single input feature or group of input features towards the predictive model's accuracy [14, 20]. This approach, in essence, transforms the task of producing feature importance estimates for a given predictive model into a supervised learning task that we can address with existing supervised machine-learning models.

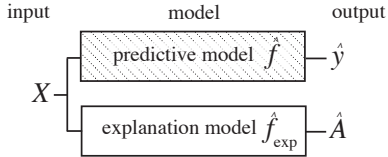

Figure 1: CXPlain trains an explanation model $\hat{f}_{\exp}$ (bottom) to learn to estimate importance scores $\hat{A}$ for a predictive target model $\hat{f}$ (top) given features $X$.

**Causal Objective.**  The core component of CXPlain is the causal objective that enables us to optimise explanation models to learn to explain another predictive model. The causal objective we build on was first introduced to jointly learn to produce accurate predictions and estimates of feature importance in a single neural network model [14]. However, the original formulation of the causal objective required a specific attentive mixture of experts architecture. In this work, we contribute an adapted version of the causal objective from [14] that does not require a specific model structure, and that can be used to train explanation models to learn to explain any machine-learning model. The causal objective introduced in [14] was based on the Humean definition of causality used by Granger [36], who defined a causal relationship $x_i \to \hat{y}$ between random variables $x_i$ and $\hat{y}$ to exist if we are better able to predict $\hat{y}$ using all available information than if the information apart from $x_i$ had been used [14]. i.e. if the absence of $x_i$ as a feature decreases our ability to predict $\hat{y}$. Granger [36]'s definition of causality was based on two key assumptions: (1) That our set of available variables $X$ contains *all* relevant variables for the causal problem being modelled, and (2) that $x_i$ temporally precedes $\hat{y}$ [36]. In the general setting, these assumptions can not be verified from observational data [37]. However, in our specific setting, we know a priori that the inputs of the predictive model $\hat{f}$ mathematically always precede its output, and that the explained model's output, on deterministic hardware and software, is not influenced by variables other than those present in its set of input features. We can therefore use the given definition to quantify the degree to which an input feature caused a marginal improvement in the predictive performance of the predictive model

$\hat{f}$. Given input covariates $X$, we therefore denote $\varepsilon_{X \setminus \{i\}}$ as the predictive model's error without including any information from the $i$th input feature and $\varepsilon_X$ as the predictive model's error when considering all available input features. To calculate $\varepsilon_{X \setminus \{i\}}$ and $\varepsilon_X$, we first compute the outputs $\hat{y}_{X \setminus \{i\}}$ and $\hat{y}_X$ of the predictive model $\hat{f}$ without and with the $i$th input feature $x_i$, respectively:

$$\hat{y}_{X \setminus \{i\}} = \hat{f}(X \setminus \{i\}) \qquad (1) \qquad\qquad \hat{y}_X = \hat{f}(X) \qquad (2)$$

There are several different approaches to obtaining $X \setminus \{i\}$ from the full set of input features, depending on the type of input data. For most types of data, masking the respective input feature $x_i$ at index $i$ with zeroes, when the zero value has no special meaning, or replacing it with the mean value across the entire data set are both valid choices [20, 21, 24]. More sophisticated feature masking schemes that consider the masked feature's distribution [38, 39] could be a more principled alternative to masking with point-wise estimates. Given $X \setminus \{i\}$, we compare the predictions $\hat{y}_{X \setminus \{i\}}$ and $\hat{y}_X$ with the ground-truth labels $y$ using the predictive model's loss function $\mathcal{L}$ to calculate $\varepsilon_{X \setminus \{i\}}$ and $\varepsilon_X$:

$$\varepsilon_{X \setminus \{i\}} = \mathcal{L}(y, \hat{y}_{X \setminus \{i\}}) \qquad (3) \qquad\qquad \varepsilon_X = \mathcal{L}(y, \hat{y}_X) \qquad (4)$$

Following Granger [36]'s definition of causality, we define the degree $\Delta \varepsilon_i$ to which the $i$th input feature causally contributed to the predictive model's output $\hat{y}$ as the decrease in error, as measured by its loss $\mathcal{L}$, associated with adding that feature to the set of available information sources [14]:

$$\Delta \varepsilon_{X,i} = \varepsilon_{X \setminus \{i\}} - \varepsilon_X \qquad (5)$$

Lastly, we normalise the importance scores $\omega_i$ to relative contributions $\in [0, 1]$ with $\Sigma_i \omega_i = 1$ [14]:

$$\omega_i(X) = \frac{\Delta \varepsilon_{X,i}}{\sum_{j=0}^{p-1} \Delta \varepsilon_{X,j}} \qquad (6)$$

We then arrive at our causal objective $\mathcal{L}_{\text{causal}} = \frac{1}{N} \sum_{l=0}^{N-1} \text{KL}(\Omega_{X_l}, \hat{A}_{X_l})$ [14] that aims to minimise the Kullback-Leibler (KL) divergence [40] between the target importance distribution $\Omega$ with $\Omega(i) = \omega_i(X)$ for a given sample $X$, and the distribution of importance scores $\hat{A}$ with $\hat{A}(i) = \hat{a}_i$ as estimated by $\hat{f}_{\text{exp}}$ based on $X$. Using $\mathcal{L}_{\text{causal}}$, we can train supervised learning models to learn to explain any other machine-learning model based solely on its outputs, and without the need to retrain the model to be explained. Precomputing the importances $\Omega$ for each training sample $X$ takes $N(p+1)$ evaluations of the target predictive model at training time. For high-dimensional images, it is sensible to group non-overlapping regions of adjacent pixels into feature groups, since removing single pixels in high-dimensional images is unlikely to strongly affect a predictive model's output [21]. This also significantly limits the number of feature groups $p$ for which importances $\omega_i$ have to be precomputed. We note that estimating $\hat{A}$ is not necessary in situations in which ground truth labels are readily available, e.g. during model development. In those situations, $\Omega$ can directly be used to explain $\hat{f}$.

**Explanation Models.** In principle, any supervised machine learning model that can be trained with a custom objective could be used as a causal explanation model. In this work, we focus on neural explanation models. Using deep neural networks as causal explanation models has the advantage that these models are able to extract high-level feature representations from high-dimensional and unstructured data [41], and thus remove the need to perform manual feature engineering. We leave the exploration of other classes of explanation models to future work. A priori, it is not clear which architectures would be most suitable to be used in neural explanation models. Absent any prior knowledge about the structure of the input data, multilayer perceptrons (MLPs) are likely a sensible default choice. However, since architectures that exploit the spatial or temporal structure of input data have been shown to be efficacious, we reason that, depending on the data modality of the input features of the model to be explained, special-purpose architectures, such as convolutional neural networks [42] for images and attentive neural networks for texts [43], could perform better than MLPs. In particular, U-nets [44] that have been designed for image segmentation, a task that involves mapping input pixels to segmentation labels, may perform well as causal explanation models for images since segmentation is semantically similar to explanation, which involves mapping input pixels to importance scores. To determine whether or not specialised model architectures can achieve better performances in neural explanation models, we experimentally evaluate both MLPs and U-nets.

**Uncertainty of Importance Estimates.** In addition to producing accurate estimates of feature importance, we wish to provide uncertainty estimates $u_i$ that quantify the uncertainty associated with each individual feature importance estimate $\hat{a}_i$ produced by a CXPlain model. In particular, we

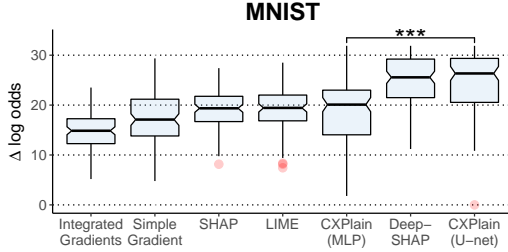
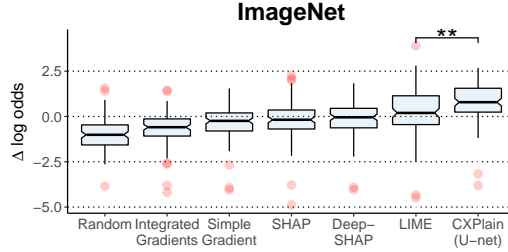

Figure 2: Comparison of the distributions of the changes in log odds $\Delta$log-odds after masking the top $10\%$ most important pixels according to several feature importance estimation methods across $N = 100$ MNIST test images (higher is better). *** = significantly different ($p < 0.001$, MWW).

Figure 3: Comparison of the distributions of the changes in log odds $\Delta$log-odds after masking the top $30\%$ most important pixels according to several feature importance estimation methods across $N = 100$ test ImageNet images (higher is better). ** = significantly different ($p < 0.01$, MWW).

would like to calculate confidence intervals $\text{CI}_{i,\gamma} = [c_{i,\frac{\alpha}{2}}, c_{i,1-\frac{\alpha}{2}}]$ with lower bounds $c_{i,\frac{\alpha}{2}}$ and upper bounds $c_{i,1-\frac{\alpha}{2}}$ at confidence level $\gamma = 1 - \alpha$ for each assigned feature importance estimate $\hat{a}_i$. The width $u_i = c_{i,1-\frac{\alpha}{2}} - c_{i,\frac{\alpha}{2}}$ of $\text{CI}_{i,\gamma}$ can subsequently be used to quantify the uncertainty of $\hat{a}_i$. To derive uncertainty estimates for causal explanation models, we propose the use of bootstrap ensemble methods, specifically using bootstrap resampling [45, 46]. To train bootstrap ensembles of causal explanation models, we first draw $N$ training samples $X$ at random with repeats from the original training set. We then train an explanation model using the before-mentioned causal objective until convergence on the selected subset of the training set. We repeat this process $M$ times to obtain a bootstrap ensemble of $M$ explanation models (Algorithm in Appendix B). We use the median of the attributions $\hat{a}_i$ of the ensemble members as the assigned importance of the bootstrap ensemble, and the $\frac{\alpha}{2}$ and $1 - \frac{\alpha}{2}$ quantiles as lower and upper bounds of its CI, respectively. The efficacy of bootstrap ensembles for estimating the uncertainty in outputs of neural networks has been demonstrated in, e.g., [47], but this work is, to the best of our knowledge, the first to consider using bootstrap ensembles of explanation models to quantify the uncertainty in assigned importance scores. We note that Monte Carlo dropout [48], which uses dropout [49] at evaluation time, is an alternative method for estimating uncertainty for the outputs of neural networks that does not require explicitly training an ensemble of models, but may not always produce uncertainty estimates of the same quality as ensembles [47].

## 4 Experiments

Our experiments aimed to answer the following questions:

1 How does the feature importance estimation performance of CXPlain compare to that of existing state-of-the-art methods?
2 How does the computational performance of CXPlain compare to existing model-agnostic and model-specific methods for feature importance estimation?
3 Are uncertainty estimates computed via bootstrap resampling of CXPlain models qualitatively and quantitatively correlated with their ability to accurately determine feature importance?

To answer these questions, we performed extensive experiments on several benchmarks that compare both the computational as well as the estimation performance of CXPlain to existing state-of-the-art methods for feature importance estimation. To enable a meaningful comparison, we focus most of our experiments on image classification tasks, where we are best able to visualise and quantify the performance of feature importance estimation methods, and on neural network models as models to be explained, since most existing model-specific attribution methods that we wish to compare to were developed exclusively for neural networks. However, we note that CXPlain as a method is compatible with any machine-learning model, data modality, and both regression as well as classification tasks. We used Mann–Whitney–Wilcoxon (MWW) tests [50] to calculate $p$-values for the main comparisons.

### 4.1 Determining Important Features in MNIST and ImageNet

To compare the accuracy of CXPlain to existing state-of-the-art methods for feature importance estimation, we evaluated its ability to identify important features in MNIST [51] and ImageNet [52] images. To do so, we followed the experimental design first proposed by Shrikumar et al. [1], and

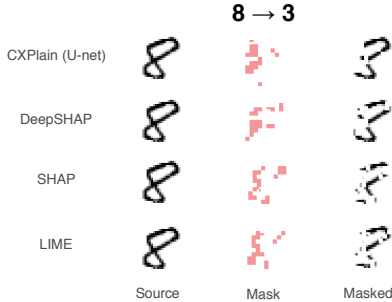

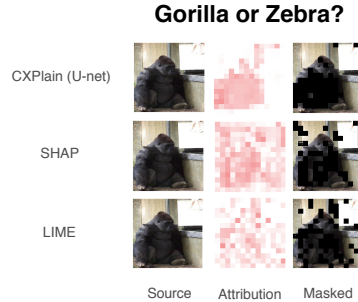

Figure 4: A comparison of the top $10\%$ most important pixels (= Mask) as identified by CXPlain (U-net), DeepSHAP, SHAP, and LIME on the same sample test set image (Source) of the 8 vs. 3 MNIST benchmark. With accurate estimates, the Masked image should more closely resemble a 3 than an 8, since the pixels that most distinguished an 8 as an 8 should have been removed.

Figure 5: A comparison of the feature importance scores (= Attribution) as estimated by CXPlain (U-net), SHAP, and LIME on the same sample test set image (Source) of the Gorilla vs. Zebra ImageNet benchmark. We found that CXPlain (U-net) produces attribution maps that are, subjectively and qualitatively, more semantically focused on the most salient regions of the image.

trained binary classification models to distinguish between two digit types (8 vs. 3) on MNIST (model accuracy: $99.85\%$), and two object categories (Gorilla vs. Zebra) on ImageNet (model accuracy: $96.73\%$). As a preprocessing step, pixel values were scaled to be in the range of $[0, 1]$ prior to training. We then used several importance estimation methods to determine which input pixels were most important for the classification models' decisions on $N = 100$ test images. We masked the top 10 and $30\%$ of those most important pixels for MNIST and ImageNet, respectively, and measured the resulting change in the classification models' confidences by computing the difference in log odds

$$\Delta\text{log-odds} = \text{log-odds}(p_{\text{original}}) - \text{log-odds}(p_{\text{masked}}) \qquad (7)$$

where $\text{log-odds}(p) = \log(\frac{p}{1-p})$, and $p_{\text{original}}$ and $p_{\text{masked}}$ are the classification models' outputs $p \in [0, 1]$ for the original image and the masked image with the top pixels removed, respectively. To ensure that the explanations $e_i$ of all methods are on the same scale, we normalised them to the range of $[0, 1]$ using the transformation $\hat{a}_i = |e_i|/\Sigma_{i=0}^{N}|e_i|$. We plotted the assigned importances and the resulting masked images to qualitatively assess each methods' ability to determine the salient features in the original image (Figures 4 and 5). We additionally recorded the mean and standard deviation of the time taken (in seconds) to compute the feature importance estimates for each method on the same hardware (Appendix C) over 10 and 5 runs with the same parameters and random seed for MNIST and ImageNet, respectively (Figures 6 and 7). Further training details are given in Appendix A.

## 4.2 Quantifying Uncertainty in Estimates of Feature Importance

To quantitatively and qualitatively assess the accuracy of the uncertainty estimates provided by bootstrap ensembles of CXPlain models, we analysed whether their uncertainty estimates $u_i$ are correlated with their errors in feature importance estimation on held-out MNIST test samples. We evaluated several numbers $M$ of bootstrap resampled models in order to determine how the number of ensemble members affects the uncertainty estimation performance of bootstrap ensembles of CXPlain models. In addition, we also evaluated the performance of randomly selected uncertainty estimates as a baseline for comparison. In general settings, it is difficult to evaluate uncertainty estimates for feature importance estimation methods, since we typically do not have per-feature ground-truth attributions to evaluate against. However, by comparing the ranking implied by the ground-truth change in log-odds to the ranking implied by the explanation model we are able to define a rank error $\text{RE}_i$ for each $x_i$. Formally, the rank error $\text{RE}_i = |\text{rank}_{\Delta\text{log-odds}}(i) - \text{rank}_{\hat{f}_{\text{exp}}}(i)|$ is the difference in rank between the true $\text{rank}_{\Delta\text{log-odds}}$ implied by $\Delta\text{log-odds}$, and the estimated $\text{rank}_{\hat{f}_{\text{exp}}}$ implied by the explanation model, where $\text{rank}_b(i)$ defines the rank of $x_i$ from 0 to $p-1$ implied by $b$.

As correlation metric, we used Pearson's $\rho$ to measure the correlation between the rank error $\text{RE}_i$ and the uncertainty estimates $u_i = c_{i,95\%} - c_{i,5\%}$ defined by the bootstrap resampled $\gamma = 90\%$ CIs for each importance estimate $\hat{a}_i$ in the top $2.5\%$ of pixels by $\Delta\text{log-odds}$ across $N = 100$ unseen images from the MNIST test set. We limited the evaluation to all pixels with a $\Delta\text{log-odds}$ greater than 0.

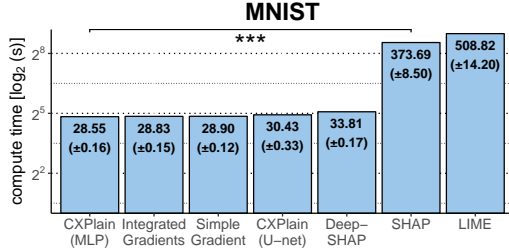
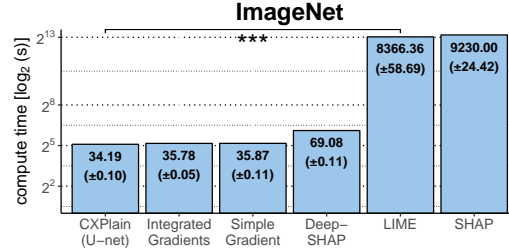

Figure 6: A comparison of the compute time (in $\log_2$ seconds) needed to produce feature importance estimates using several state-of-the-art feature importance estimation methods on the same hardware for the same $N = 100$ sample test images from the MNIST benchmark (lower is better). *** = significantly different ($p < 0.001$, MWW)

Figure 7: A comparison of the compute time (in $\log_2$ seconds) needed to produce feature importance estimates using state-of-the-art feature importance estimation methods on the same hardware for the same $N = 100$ sample test images from the ImageNet benchmark (lower is better). *** = significantly different ($p < 0.001$, MWW)

If our uncertainty estimates are well calibrated, we would expect to see a high correlation between the uncertainty estimates $u_i$ and the magnitude of rank errors $\text{RE}_i$, since that would indicate that the uncertainty estimates $u_i$ accurately quantify how certain the feature importance estimates $\hat{a}_i$ are on previously unseen sample images. For the comparison of the resulting distributions of correlation scores, we applied the Fisher z-transform to the correlation scores in order to correct for the skew in the distribution of the sample correlation [53]. Figure 9 depicts visualisations of the calculated ground-truth log odds, the rank errors of the explanation model's importance estimates, and the uncertainty for each importance estimate for three test set images. We used the same hyperparameters as in the previous experiment to train the ensembled CXPlain (MLP) models (Appendix A).

## 5   Results and Discussion

**Predictive Performance.**   We found that, on the MNIST benchmark, CXPlain (U-net) was competitive with the best competing state-of-the-art feature importance estimation method, DeepSHAP. We also found that CXPlain (U-net) produced significantly ($p < 0.001$, MWW) more accurate feature importance estimates than CXPlain (MLP) - indicating that model architectures specifically tailored for the image domain are more effective than MLPs in neural explanation models (Figure 2). On the ImageNet benchmark, CXPlain significantly ($p < 0.01$, MWW) outperformed the best competing feature importance estimation method, LIME (Figure 3). We also found that the model-specific attribution methods Simple Gradient and Integrated Gradients performed relatively poorly across both benchmarks, and were consistently outperformed by the model-agnostic attribution methods, CXPlain, and DeepSHAP. Qualitatively, we found that the estimates of feature importance provided by CXPlain were more focused on the subjectively more important semantic regions of the sample images from both MNIST and ImageNet (Figures 4 and 5; more in Appendix D). Other methods, in contrast, produced more superfluous attributions. This behavior is exhibited in Figure 5 where SHAP and LIME both attribute significant importance to the wall behind the gorilla, whereas CXPlain focused nearly all its attention on the gorilla itself, with the exception of the window frame receiving some importance outside the top 30% of importances of that sample image. We believe this could be due to the fact that the causal objective strongly penalises attributions outside regions of interest - leading to qualitatively more focused estimates of importance.

**Computational Performance.**   In terms of computational performance, we found that CXPlain computed feature importance estimates significantly faster than the state-of-the-art model-agnostic attribution methods, LIME and SHAP, on both the MNIST and ImageNet benchmarks (Figures 6 and 7). Gradient-based attribution methods and CXPlain performed similarly. On ImageNet, the gap between LIME and SHAP and the faster methods was considerably larger than on MNIST, since the large numbers of model evaluations for LIME and SHAP were slower on higher-dimensional images.

**Quality of Uncertainty Estimates.**   We found that, quantitatively, even relatively small CXPlain ensembles with just $M = 5$ bootstrap resampled models produce uncertainty estimates that are significantly ($p < 0.001$, MWW, compared to Random) correlated with its ability to accurately estimate feature importances on $N = 100$ previously unseen test images (Figure 8). We also found

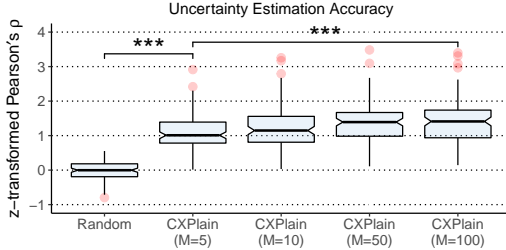
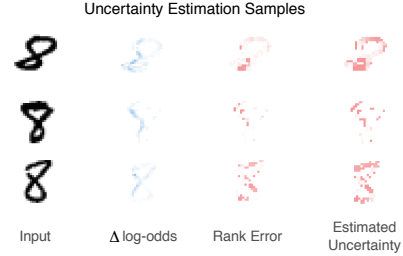

Figure 8: A comparison of the distributions of the z-transformed Pearson's correlations $\rho$ between the uncertainty estimates $u_i$ produced by various numbers $M$ of bootstrapped ensembles of CX-Plain models and the Random baseline, and the ground-truth rank errors of the top $2.5\%$ most important pixels across $N = 100$ unseen test images from the MNIST benchmark (higher is better). *** = significantly different ($p < 0.001$, MWW)

Figure 9: Visualisations of the calculated ground-truth change in log odds $\Delta$log-odds, the Rank Errors of the explanation model's feature importance estimates, and the Estimated Uncertainty for each feature importance estimate as obtained via bootstrap resampling ($M = 100$) for three unseen sample test set images (Input) from the MNIST benchmark. Note the visual similarity of the Rank Error and the Estimated Uncertainty.

that increasing the size $M$ of the bootstrap ensemble further significantly ($p < 0.001$ for $M = 5$ to $M = 100$, MWW) increases this correlation, and, thus, the quality of the provided uncertainty estimates. Qualitatively, there was a high visual similarity between the uncertainty estimates $u_i$ provided by the CXPlain ensembles for each input feature $x_i$ and the magnitude of rank errors $\text{RE}_i$ committed by its importance estimates $\hat{a}_i$ (Figure 9). The large differences in importance estimation accuracy between state-of-the-art feature importance estimation methods shown in the MNIST and ImageNet benchmarks indicate that many of the importance estimates they provide are not truthful to the predictive model $\hat{f}$ to be explained, and that measures of uncertainty are necessary to fully understand the expected reliability of feature importance estimates.

**Limitations.** While they are fast at evaluation time, a limitation of CXPlain models is that they have to be trained to learn to explain a predictive model. However, this one-off compute cost typically amortises quickly, since CXPlain is significantly faster at evaluation time than existing model-agnostic importance estimation methods. Another important point to note is that the associations identified by CXPlain models are only causal in the sense that they quantify the degree to which the input features $x_i$ caused a marginal improvement in the predictive performance of the predictive model $\hat{f}$. Associations reported by CXPlain, in particular, do not in any way indicate that there is a causal relationship between the explained model's input and output variables in the real world.

## 6  Conclusion

We presented CXPlain, a new method for learning to estimate feature importance for any machine-learning model. CXPlain is based on the idea of training a separate explanation model to learn to estimate which features are important for a given output of a target predictive model using a causal objective. This approach has several advantages over existing ones: It is compatible with any machine-learning model, can produce estimates of feature importance quickly after training, and may be combined with bootstrap resampling to obtain uncertainty estimates for the provided feature importance scores. We showed experimentally that CXPlain is significantly more accurate in estimating feature importance than existing model-agnostic methods on both MNIST and ImageNet benchmarks, while being orders of magnitude faster at providing importance estimates than state-of-the-art model-agnostic methods. We also found that, analogous to standard supervised learning tasks, special-purpose model architectures may improve the performance of neural explanation models in images, and that the bootstrap resampled uncertainty estimates for the importance scores of an explanation model are significantly correlated with CXPlain's ability to accurately estimate feature importance - indicating that bootstrap resampling is a suitable approach for quantifying the uncertainty of importance estimates. Causal explanation models that both produce accurate estimates of feature importance and their uncertainties quickly for any machine-learning model and data modality may enable users to better understand, validate, and interpret machine-learning models, while also informing them when their explanations can not be expected to be accurate.

**Acknowledgments**

This work was partially funded by the Swiss National Science Foundation (SNSF) project No. 167302 within the National Research Program (NRP) 75 "Big Data". We gratefully acknowledge the support of NVIDIA Corporation with the donation of the Titan Xp GPUs used for this research. Patrick Schwab is an affiliated PhD fellow at the Max Planck ETH Center for Learning Systems. We additionally thank the anonymous reviewers whose comments helped improve this manuscript.

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
