[Supplementary Material]

# Supplementary Material for: "CXPlain: Causal Explanations for Model Interpretation under Uncertainty"

**Patrick Schwab and Walter Karlen**
Institute of Robotics and Intelligent Systems, ETH Zurich
`patrick.schwab@hest.ethz.ch`

## A   Hyperparameters

We implemented all our experiments using Python and TensorFlow [52], and used standardised compute hardware to run the experiments (see Appendix C). For both benchmarks, we used 10000 perturbed samples per explained image for both LIME and SHAP. The output layer of all CXPlain models had one output node for each of the $p$ output feature importance scores $a_i$, and was followed by a softmax activation. We used reference implementations provided by the method's original authors for LIME[1], SHAP and DeepSHAP[2], and our own implementations for Simple Gradient and Integrated Gradients. To keep the computation time for sensitivity-based attribution methods in a reasonable range, we explain non-overlapping connected regions of 2x2 pixels for the MNIST benchmarks, and regions of 16x16 pixels for the ImageNet benchmarks. Since the image dimensions were 28x28 for MNIST and 224x224 for ImageNet, the target attribution maps were of size 14x14 for both benchmarks. To keep the comparison meaningful, we summed and renormalised the attributions for each block for the methods that produced per-pixel importances to downsample their attribution maps to the same resolution. For CXPlain, we also used a target attribution map size of 14x14. For the ImageNet benchmark, we split the dataset at random stratified by class into training (60%), validation (20%), and test set (20% of all samples). For MNIST, we used the splits from [49].

**MNIST Benchmark.**   As target predictive model, we used a binary classifier ResNet-20 model using rectified linear (ReLu) units without batch normalisation [53] that was trained with the Adam [54] optimiser, a learning rate of 0.001, weight decay of 0.001, a batch size of 32, for a maximum of 50 epochs and an early stopping patience of 12 on the validation set loss, and achieved a test set accuracy of $99.85\%$ in distinguishing between the two digit classes. The CXPlain (MLP) model was trained with the Adam [54] optimiser, a learning rate of 0.001, a batch size of 100, 2 hidden layers, $H$ hidden units per hidden layer each followed by a scaled exponential linear unit (SELU) activation [55], for a maximum of 50 epochs with an early stopping patience of 12 on the validation set loss. The CXPlain (U-net) model was trained with the Adam [54] optimiser for a maximum of 50 epochs with an early stopping patience of 12 on the validation set loss, a learning rate of 0.002, a batch size of 128, $K$ filters in the first convolutional layer and $K * 2^{\text{layer\_index}}$ filters in every subsequent pair of convolutional layer for a maximum of 2 pairs of convolutional layers in the first stage of the U-net followed by a max pooling operation that reduced the dimensionality of the layer input both in width and height in half. The same steps were then mirrored in the inverse direction as outlined in [41] until the target attribution map dimension of $14 * 14$ was reached. Each convolutional layer was followed by a ReLu activation. We used a total of 5 hyperparameter optimisation runs on the validation set to select the number $H$ of hidden units per hidden layer of the CXPlain (MLP) model, the number $K$ of initial convolutional filters of the CXPlain (U-net) model, and the dropout probability of both at random from predefined ranges (Tables S1 and S2). The CXPlain (MLP) model selected after

hyperparameter optimisation used a dropout rate of $4.01\%$, 126 hidden units per hidden layer, and was trained in 288.73 seconds after precomputing $\Omega$ for each sample $X$. The CXPlain (U-net) model selected after hyperparameter optimisation used a dropout rate of $0.001\%$, 77 initial convolutional filters, and was trained in 499.38 seconds after precomputing $\Omega$ for each sample $X$. We used the digits 8 and 3 for the MNIST benchmark.

**ImageNet Benchmark.**   As target predictive model, we used a binary classifier ResNet-32 model using rectified linear (ReLu) units without batch normalisation [53] that was trained with the Adam [54] optimiser, a learning rate of 0.01, a batch size of 32, for a maximum of 250 epochs and an early stopping patience of 12 on the validation set loss, and achieved a test set accuracy of $96.73\%$ in distinguishing between the two object classes. During training, we used automated data augmentation that transformed the image with a randomised shear, zoom, width shift, and height shift of up to 10%, rotated the image at most 20 degrees and flipped the images horizontally at random. The CXPlain (U-net) model was trained with the Adam [54] optimiser, a learning rate of 0.001, a batch size of 32, $K$ filters in the first convolutional layer and $K * 2^{\text{layer\_index}}$ filters in every subsequent pair of convolutional layer for a maximum of 5 pairs of convolutional layers in the first stage of the U-net followed by a max pooling operation that reduced the dimensionality of the layer input both in width and height in half. The same steps were then mirrored in the inverse direction as outlined in [41]. Each convolutional layer was followed by a ReLu activation. We used a total of 5 hyperparameter optimisation runs on the validation set to select the number $K$ of initial convolutional filters and dropout probability of the CXPlain (U-net) model at random from predefined ranges (Table S3). The CXPlain (U-net) model selected after hyperparameter optimisation used a dropout rate of $5.61\%$, 12 initial convolutional filters, and was trained in 372.14 seconds after precomputing $\Omega$ for each sample $X$. The ImageNet synsets we used for the benchmark were zebra (n02391049) and gorilla (n02480855).

**Twitter Sentiment Analysis Benchmark.**   In addition to the benchmarks presented in the main body of the paper, we also performed qualitative experiments using a sentiment analysis model for short text messages in order to demonstrate the efficacy of CXPlain for data modalities other than images, and target predictive models other than neural networks. As training dataset, we used a random subset of $N = 100000$ short messages (50000 positive and 50000 negative messages) from the dataset available at `http://cs.stanford.edu/people/alecmgo/trainingandtestdata.zip`. Like the ImageNet benchmark, we split the dataset at random stratified by class into training (60%), validation (20%), and test set (20% of all samples). We then trained a random forest (RF) classifier with 64 trees to classify short messages as being either positive or negative in content as our target predictive model. The model achieved a test set accuracy of $76.32\%$. The RF model received word count vectors over a vocabulary initialised with the training set as inputs. The input text was lowercased, punctuation was removed, and the words were preprocessed using the Natural Language Toolkit (NLTK) tokeniser available at `https://www.nltk.org/`. As explanation model, we trained a CXPlain (MLP) model that received a fixed length sequence of 96 word IDs according to the previously mentioned vocabulary in order to determine which words were most important for the RFs outputs. Messages shorter than 96 were padded with the zero ID that was not assigned to any other words, and words that were not in the training vocabulary were assigned an ID representing unknown words that was not assigned to any other words. The CXPlain (MLP) used an initial embedding layer to transform the word IDs into an embedding space that was followed by a number of $L$ hidden layers with $H$ hidden units each. Each layer was followed by a SeLU activation [55]. The CXPlain (MLP) model was trained with the Adam [54] optimiser, a learning rate of 0.0001, a batch size of 128, and a dropout percentage $p_{\text{dropout}}$ for a maximum of 100 epochs and an early stopping patience of 12 on the validation set loss. $H$, $L$ and $p_{\text{dropout}}$ were selected at random from predefined ranges over 5 hyperparameter optimisation runs using the lowest validation loss as the selection criterium (Table S4). The CXPlain (MLP) model selected after hyperparameter optimisation used a dropout rate of $5.47\%$, 162 hidden units per hidden layer, 1 hidden layer, and was trained in 126.28 seconds after precomputing $\Omega$ for each sample $X$. To remove the information from the $i$th word $x_i$ for the calculation of the causal objective, we simply deleted the respective word from the sentence. See Appendix E for qualitative samples of importances assigned by the selected CXPlain (MLP) to short text messages from the Twitter Sentiment Analysis benchmark.

## B  Training Bootstrap Ensembles of Causal Explanation Models

---

**Algorithm 1** Training Bootstrap Ensembles of Causal Explanation Models.

---

**Input:**
 1: Training dataset $T$ consisting of $N$ samples $X$ with ground-truth labels $y$
 2: Size $M$ of ensemble
 3: Target predictive model $\hat{f}$ to be explained
**Output:** Ensemble $E$ of $M$ causal explanation models
 4: **procedure** TRAIN_EXPLANATION_ENSEMBLE:
 5:     $E \leftarrow$ Empty
 6:     **for** $i$ from 0 to $M - 1$ **do**
 7:         $T_{\text{subset}} \leftarrow$ Draw $N$ pairs of samples $(X, y)$ at random with repeats from $T$
 8:         Train explanation model CXPlain$_i$ until convergence using $\mathcal{L}_{\text{causal}}$ with $\hat{f}$ and $T_{\text{subset}}$.
 9:         Add CXPlain$_i$ to $E$
         **return** E

---

## C  Computing Infrastructure

We used the same hardware for all experiments: Intel Core i5 7600K, Nvidia GeForce Titan Xp, 32 GB RAM.

## D  Qualitative Samples for the MNIST and ImageNet Benchmarks

We present more qualitative samples from the MNIST benchmark in Figure S1, and more qualitative samples from the ImageNet benchmark in Figure S2.

## E  Qualitative Samples for the Twitter Sentiment Analysis Benchmark

We show qualitative samples of the importances assigned to short messages in the Twitter Sentiment Analysis benchmark by the CXPlain (MLP) in Table S5. We found that, qualitatively, the explanations of CXPlain (MLP) provided for the RF were indeed high for words that have positive or negative connotations, and, subjectively, appeared to be semantically meaningful.

Table S1: Hyperparameter ranges used to train CXPlain (MLP) in the MNIST benchmark.

| Hyperparameter | Values |
| --- | --- |
| Number of hidden units per hidden layer $H$ | $[70, 140]$ |
| Dropout percentage $p_{\text{dropout}}$ | $[0\%, 10\%]$ |

Table S2: Hyperparameter ranges used to train CXPlain (U-net) in the MNIST benchmark.

| Hyperparameter | Values |
| --- | --- |
| Number of initial convolutional filters $K$ | $[65, 80]$ |
| Dropout percentage $p_{\text{dropout}}$ | $[0\%, 10\%]$ |

Table S3: Hyperparameter ranges used to train CXPlain (U-net) in the ImageNet benchmark.

| Hyperparameter | Values |
| --- | --- |
| Number of initial convolutional filters $K$ | $[8, 24]$ |
| Dropout percentage $p_{\text{dropout}}$ | $[0\%, 10\%]$ |

Table S4: Hyperparameter ranges used to train CXPlain (MLP) in the Sentiment Analysis benchmark.

| Hyperparameter | Values |
| --- | --- |
| Number of hidden units per hidden layer $H$ | $[64, 180]$ |
| Number of hidden layers $L$ | $[1, 3]$ |
| Dropout percentage $p_{\text{dropout}}$ | $[0\%, 10\%]$ |

Figure S1: Additional qualitative comparisons of the top 10% most important pixels (= Mask) as identified by CXPlain (U-net), DeepSHAP, SHAP, and LIME on two sample test set images (Source) of the 8 vs. 3 MNIST benchmark.

# Gorilla or Zebra?

CXPlain (U-net)

SHAP

LIME

Source   Attribution   Masked

# Gorilla or Zebra?

CXPlain (U-net)

SHAP

LIME

Source   Attribution   Masked

Figure S2: Additional qualitative comparisons of feature importance scores (= Attribution) as estimated by CXPlain (U-net), SHAP, and LIME on two same sample test set images (Source) of the Gorilla vs. Zebra ImageNet benchmark.

Table S5: Examples of short messages and the importances assigned by CXPlain (MLP) in the Twitter Sentiment Analysis benchmark. Deeper colors indicate higher importances, as indicated in the labelled examples in the header row of the table. All samples are labelled as positive in sentiment.

| Short messages | highest importance | lowest importance |
| --- | --- | --- |

you re welcome glad you enjoyed it

is awesome thanks got some good lolz needed it xxx

today its already busy i wish it was slow maybe later hopefully

whooooooo finally done with high school thank god just graduation now yay

happy emox awe i d be sad if you bear napped him

thank you all so much for your kindness

bout half way done packing gon na be a long ride thanks for the sketch hanna

thanks for all the gr follows in the last hrs i m awed with each of you

what are yall doing in the lb you guys should kick it at my place

looking forward to his birthday tomorrow

## Footnotes

[1]https://github.com/marcotcr/lime

[2]https://github.com/slundberg/shap