[Reviews · NeurIPS 2019]

Reviewer 1



I’m not very familiar with recent approaches to explanation methods in ML, therefore my low confidence. 1) I am not entirely convinced that an amortized explanation model is a reasonable thing to consider; this could possibly stem from my incomplete understanding of the major use cases of an explanation model. I imagine it to be most useful in practice to investigate outliers / failure cases of the system in question. If this is correct, the explanation method does not necessarily need to be very fast, as it’s only used in rare failure cases. Furthermore, (under this use case assumption) the explanation model would only be useful if it matches the true model on (catastrophic) miss-classification examples; this is inherently hard to guarantee if this is not taken into account during the design and training of the explanation model. A straight-forward computation of feature importance on the ground truth system would be more appropriate in this case I presume. 2) Eqn (5): This definition of feature importance only considers contributions of features towards the ground truth label. It does not attribute importance to features that potentially massively change how the model goes wrong, as long as the error stays the same; eg on ImageNet it would not assign importance to pixels that change the prediction from one wrong class to another, although intuitively this could be an interesting piece of information to debug a classifier. Could the authors comment on this choice? 3) I might be misunderstanding the evaluation of the feature importance uncertainty in section 4.2. Why can the authors not compute the ground truth feature importance on the test set and check is the predictive uncertainty is well calibrated to these held-out values? Ie why is the rank-based method necessary?

Reviewer 2



* Update after author response* Thanks to the authors for their thoughtful and detailed responses. Most of the responses were quite clarifying (esp. in regards to noting that there is a separate Omega for each x!) and I'll increase my score from a (4) to a (5). I still think much more clarity is needed in describing the methodology, the overall goal, and defining carefully what they mean by "causal." Other referees noted that the interest in this paper is Granger causality and not in understanding what might have happened under intervention (the authors mention this as well in their response, noting that "our goal is not to estimate what would happen if a particular feature's value changed"). In the present form, I worry that most readers will think the paper is about this second notion and get quite confused. ------- The authors set a rather ambitious goal for themselves: to produce fast and accurate estimates of variable importance that can be applied to *any* machine learning model. They do this by framing the question causally: i.e. identify which inputs causally affect the outputs of a machine learning model. They note, importantly, that their procedure does not generally provide any information about the causal mechanisms that generated the data. All it does is provide an explanation for a trained model's predictions. That being said, I am not sure that I completely understood the specifics of the proposed procedure. In what follows, I'll attempt to summarize my understanding and I would greatly appreciate any clarification from the authors. The authors begin by assuming that one has pairs of (x_n, y_n) of covariate x_n and observations y_n and that one has access to a prediction machine $\hat{f}.$ Further, one is able to use this prediction machine to obtain $\hat{y}_{n}$ and $\hat{y}_{n,-i},$ the predicted outcome for observation $n$ given the full set of predictors $x_{n}$ and given all the predictors but X_i. Based on these two predictions, the authors define a discrepancy $\Delta \epsilon_{X,i}$ to be the difference in score/loss between the two predictions. To draw an analogy to standard linear model theory, this discrepancy plays a role similar to the partial-F statistic in linear models: it measures how much is added, from a predictive standpoint, by including a specific covariate in a model that already contains all of the others. From these discrepancies, the authors then derive a single vector of importance weights, $\Omega.$ Note that this vector is specific to the prediction model $\hat{f}$ and one can similarly define another set of importance weights for a different prediction model. As far as I understood, the central idea of CXPlain is to train an explanation model so that the corresponding set of importance weights is as close as possible to the true set of importance weights computed using the original prediction model. If so, then, it would be helpful to make this point explicit in the exposition. Additionally, can the authors clarify what is being averaged in the definition of the causal objective? If I understood correctly, the importance weights $\omega$ are computed for the entire dataset and not an individual datapoint. As such, it does not initially make any sense why we would be averaging over the entire dataset, as suggested by the notation. If this is not the case, then I would ask the authors to be a bit more precise in their notation. Notwithstanding these minor notational points, I am still confused about why one needs to do anything after computing $\Omega.$ After all, the quantity $\omega(i)$ precisely measures the relative gain in prediction from including predictor $i$ to a model that already included the other predictors. Furthermore, if the goal is to determine what might happen to our predictions if we change a particular feature slightly keeping all others fixed, I don't see any role for the explanation model -- one can simply compute the new prediction. Finally, if we take the explanation model to be exactly the original prediction model, we may trivially minimize the proposed causal objective. In light of this, I would appreciate some additional clarification about what is gained by learning A and not just reporting Omega directly. Some additional clarity on why the authors are using a KL discrepancy is merited. Why not use, say, the euclidean distance between the vector Omega and the importance weights derived from the explanation model? --- Originality: The authors note that the causal objective was first introduced in reference 14. The main contribution therefore seems to be a different architecture for the explanation model Clarity: The paper was well-written but was somewhat terse in terms of motivating the specific methodology proposed. Quality & Significance: I am unable to comment on the quality or significance as it is not clear to me why the explanation model is needed in general.

Reviewer 3



Originality: The concept of learning to explain is relatively new and unexplored. I can only think of a few references: some relatively unrelated NLP papers, the L2X paper (which the authors reference), and "TED: Teaching AI to explain its decisions", 2019 [which I do not particularly like]. However, as far as I know, the proposed methodology is novel. Quality: I really like the direction that this paper goes into, namely combining learning to explain and causality, as well as the experiments. The related work section is also fairly extensive. Still, I have a few issues with it: (a) Like many others, I am a fan of structured equation models and Pearl's theory of causality, and was a bit disappointed when I discovered deep in Section 3 that they were not used. I strongly suggest the authors to clarify that the paper is about Granger causality as soon as possible. (b) The uncertainty estimation idea seems disjoint from the main contribution; I think that the latter would have stood on its own. (c) From a more methodological perspective, I really dislike that: (1) feature contribution is determined by masking features one by one; this is essentially equivalent to assuming that feature contributions are additive, which is *very* unlikely to be the case in practice, especially for very non-linear models like deep nets. I realize that this is computationally advantageous, but it is a scientific duty to point out that it is an extreme approximation, probably worth revisiting in the future. There are also techniques that tackle similar problems in the literature, and it may be worth to point the reader in their direction, e.g.: "VOILA: Efficient Feature-value Acquisition for Classification". 2007. (2) Replacing a masked value by a point-wise estimation can be very bad, especially when the classifiers output changes based on the masked feature. Why would the average value (or, even worse, zero) be meaningful? It would have made sense to formalize this step better, for instance by replacing a point-wise estimante of the response variable with a distribution instead, like done in: "Quantifying Causal Influence", 2013 (a masked feature is replaced by its marginal distribution) "What to Expect of Classifiers? Reasoning about Logistic Regression with Missing Features", 2019. Again, it is a matter of pointing out that alternative can be conceived. (3) It would also be interesting to compare the proposed method with causal inference technique for SEMs, or at least mention them. I assume these to be more precise but also fatally slower. (4) It seems to me that the chosen performance measure may correlate much more with the Granger-causal loss than with the objectives optimized by the other explainers. This would bias the experiments. The authors should explain why this is not the case, or fix the performance measure otherwise. Clarity: The ideas are presented clearly. Significance: This paper has a lot of potential.

[Author Response · NeurIPS 2019]

**R1:** *"I am not entirely convinced that an amortized explanation model is a reasonable thing. To investigate outliers (...) I presume computation of feature importance on the ground truth system would be more appropriate."*

**R2:** *"I would appreciate some clarification about what is gained by learning $\hat{A}$ and not just reporting $\Omega$ directly."*

We thank R1, R2 and R3 for their insightful feedback. In settings where we have access to samples with associated ground truth labels, we could indeed directly use $\Omega$, as defined by Eq. (6), to explain the a predictive model without training a separate explanation model. As correctly pointed out by R1, this would be preferable, for example for debugging at model development time, because $\Omega$ can be computed without any uncertainty, and computational performance is not a major concern at that stage. However, $\Omega$ can *only* be computed given ground truth labels. For many use cases of explanation methods, ground truth labels are not available, and an explanation model that generalises beyond the training data is therefore necessary. Imagine, for example, a predictive model that indicates whether or not an individual is at high risk for heart failure based on her individual attributes $x_i$. Suppose now that this system indicates an increased risk of heart failure for a specific person. For the physician that receives this prediction, it would be paramount to know whether or not this prediction is caused by the patient's blood pressure reading or by their genomic information, as this would dramatically change which further clinical steps should be taken. In this setting, every model decision is explained, no ground truth labels are available, and explanations and their certainty are consequential. This setting is not unusual, since we would not need to train a predictive model if we readily had access to accurate labels.

**R1:** *"(The objective) does not attribute importance to features that change how the model goes wrong (...)"*

The perturbed feature itself would receive the same importance, but, since all attributions $\hat{a}_i$ are conditional on all input features $x_i$, the overall distribution $\hat{A}$ of importance scores would change along with the model's reasoning. In which case the user would be informed that the perturbation dramatically changed *how* the model arrived at its decision. This very approach was used in [19] to show that the explanations used by models are not robust to small perturbations.

**R1:** *"Why is the rank-based method necessary?"*

We chose the rank-based method to show that the uncertainty estimates reflect accuracy according to the log-odds metric that is widely accepted by the research community as a benchmark for feature importance estimation (e.g. [1, 6]). We believe this is a higher standard, and therefore stronger evidence, since the rank-based metric shows that the uncertainty estimates are accurate not just by our metric, but by the community's standard.

**R2:** *"Additionally, can the authors clarify what is being averaged in the definition of the causal objective?"*

The causal objective is averaged over all $N$ samples in the dataset. Every data point has an $\Omega$. We originally omitted the data point indices for brevity, but we will make the dependence of $\Omega$ and $\hat{A}$ on the sample explicit in the next revision.

**R2:** *"If the goal is to determine what might happen to our predictions if we change a particular feature slightly keeping all others fixed, I don't see any role for the explanation model – one can simply compute the new prediction.*

Our goal is not to estimate what would happen if a particular feature's value changed, but to provide a causal explanation for the prediction made by the model, i.e. which input features $x_i$ causally influenced the prediction and to what degree.

**R2:** *"Some additional clarity on why the authors are using a KL discrepancy is merited. Why not use, say, the euclidean distance between the vector Omega and the importance weights derived from the explanation model?"*

The KL divergence has connections to Bayesian surprise and human attention (see Itti and Baldi, NIPS 2006), and is therefore a particularly suitable candidate for optimising the distribution $\hat{A}$ of importance attributions.

**R3:** *"Masking one by one; this is essentially equivalent to assuming that feature contributions are additive."*

We do not define a feature's importance as its additive contribution to the model output, but as it's marginal reduction in prediction error. This subtle change in definition allows us to efficiently compute feature importance one by one. Non-linear interactions between model inputs and model outputs are possible within this definition, since the additivity constraint pertains to the marginal reduction in prediction error only (which holds in the general setting).

**R3:** *"Replacing a masked value by a point-wise estimation can be very bad, especially when the classifiers output changes based on the masked feature. Why would the average value (or, even worse, zero) be meaningful?"*

R3 is absolutely correct. There is a range of imputation strategies that could be employed to mask the individual features $x_i$, and our work focused on the most straight-forward strategies. We will clarify this point in the next revision.

**R3:** *"It would also be interesting to compare the proposed method with causal inference technique for SEMs."*

Recent work [29] has explored the use of SEMs for model attribution in deep learning. Compared to CXPlain, the main disadvantages of their approach were that (i) their method was limited to specific neural network architectures whereas CXPlain can explain any machine-learning model, and (ii) attribution time was considerably slower than CXPlain.

**R3:** *"It seems to me that the chosen performance measure may correlate much more with the Granger-causal loss than with the objectives optimized by the other explainers."*

Related works, such as LIME, SHAP and gradient-based methods, compute attributions directly based on the change in the explained model's output as also measured by the log-odds metric. In contrast, the causal loss uses the marginal reduction in prediction error and, therefore, only indirectly models the change in model output.

[Meta-Review · NeurIPS 2019]

This is a solid, well-supported addition to the literature on explainable methods for neural networks. Evaluations of the proposed system show faster and higher quality results than related systems. Reviewers found the author response compelling.